

# Approaches combining methods of Operational Research with Business Process Model and Notation: A systematic review

Hana Tomaskova[1] and Gerhard-Wilhelm Weber[2,3]

[1] University of Hradec Kralove, Faculty of Informatics and Management, Hradec Kralove, Czech Republic
[2] Faculty of Engineering Management, Poznan University of Technology, Poznan, Poland
[3] Institute of Applied Mathematics, Middle East Technical University, Ankara, Turkey

## ABSTRACT

**Background:** Business process modelling is increasingly used not only by the companies' management but also by scientists dealing with process models. Process modeling is seldom done without decision-making nodes, which is why operational research methods are increasingly included in the process analyses.
**Objective:** This systematic literature review aimed to provide a detailed and comprehensive description of the relevant aspects of used operational research techniques in Business Process Model and Notation (BPMN) model.
**Methods:** The Web Of Science of Clarivate Analytics was searched for 128 studies of that used operation research techniques and business process model and notation, published in English between 1 January 2004 and 18 May 2020. The inclusion criteria were as follows: Use of Operational Research methods in conjunction with the BPMN, and is available in full-text format. Articles were not excluded based on methodological quality. The background information of the included studies, as well as specific information on the used approaches, were extracted.
**Results:** In this research, thirty-six studies were included and considered. A total of 11 specific methods falling into the field of Operations Research have been identified, and their use in connection with the process model was described.
**Conclusion:** Operational research methods are a useful complement to BPMN process analysis. It serves not only to analyze the probability of the process, its economic and personnel demands but also for process reengineering.

# INTRODUCTION

It has been more than 15 years since 'Business Process Model and Notation' or 'Business Process Modelling Notation' (BPMN) became the official notation for process modelling. During its lifetime, this notation has gained many users and, thanks to its user-friendliness, it is used in many areas. This wide usage has led to the interconnection and use of other technologies and methods. The fundamental problem of any complex process is decision making. Operational Research as a popular scientific approach is so often

Corresponding author
Hana Tomaskova,
hana.tomaskova@uhk.cz

associated with procedural issues, making its connection to BPMN is more than natural. This article focuses on the analysis of the relationship between the Business Process Model and Notation (BPMN) process modelling and specific methods of Operational Research.

Business Process Modelling Notation was created by the Business Process Management Initiative (BPMI) as an open standards. It is very similar to flowcharts and Petri nets but offers much more sophisticated tools to describe and simulate behaviour. *Silver (2009)* stated that this approach is an 'event-triggered behaviour,' a description of the 'something happened' mode. Business Process Modelling is used to describe, recognize, re-engineer, or improve processes or practices, *Tomaskova (2017)*. Business Process Model and Notation (BPMN) is the language that is used to model business process steps from start to end. The notation was explicitly designed for wide-ranging use in process analysis, *The Object Management Group (2011)*. BPMN is both intelligible to non-specialists and allows a complicated processes between different participants to be represented. Another, very significant feature of BPMN is its 'business-friendly' orientation, which is essential for the company's business and knowledge.

Operational Research (OR) is concerned with formulating, modelling, and solving a variety of decision-making situations to find the optimal solutions. The company's philosophy and decide over business data are the most crucial management actions. The task of the manager is to select in the real system the problem to be analyzed and to formulate it precisely. The standard way of doing this involves the expression of the economic model and then the formulation of a mathematical model. It is necessary to build a simplified model of the real financial systems that only includes the essential elements that describe the formulated problem. The manager has to set the goal of the analysis and subsequent optimization. It is important to define all operations and processes that influence this goal, to describe all the factors, and to verbally express the relationships between the stated purpose and the mentioned processes and factors.

The article is divided into the following parts. The "Related works and background" section lists research articles that are relevant to a given combination of BPMN and OR areas and briefly. That part briefly provides essential information regarding the approaches that are fundamental to this systematic review. The "Research methodology" section describes a systematic search, i.e. entry conditions, exclusion criteria and limitations. The "Results" section presents the results of the analysis of articles fulfilling the requirements of the systematic review. We analyzed publications according to when they were published, their citations, the scientific areas covered, the cooperation of the authors and their keywords. Subsequently, we examined selected articles in terms of methodology, approach and research areas. In the "Discussion", we focus on scientific gaps and future research. We present a research area where we expect an increase in publications, including their specific components. We also discuss the future development of applied methods and approaches. Finally, the "Conclusion" section summarizes the results and benefits of this study.

## RELATED WORKS AND BACKGROUND

The background information and related works are listed in the paragraphs below. We first focused on process modelling and BPMN and then on OR and its essential methods and approaches.

Organizational processes and decision support can be captured in many ways, and for many areas, we can mention, for example: strategic management by: *Maltz & Kohli (1996)*, *Certo (2003)*, *Tomaskova (2009)*, *Maresova (2010)*, *Tsakalidis et al. (2019)*; product development research and innovation implementation, see *Repenning, 2002*, *Garcia (2005)*; IT and economic analyzes see *Shane & Cable (2002)*, *Dedrick, Gurbaxani & Kraemer (2003)*, *Krenek et al. (2014)*, *Tomaskova, Kuhnova & Kuca (2015)*, *Maresova, Tomaskova & Kuca (2016)*, *Tomaskova et al. (2016)*, *Maresova, Sobeslav & Krejcar (2017)*, *Cheng et al. (2019)*, *Tomaskova, Kopecky & Maresova (2019)*, *Tomaskova et al. (2019)*, *Kopecky & Tomaskova (2019)*, *Kopecky & Tomaskova (2020)*; different simulation approaches analysis, see *Sterman (1994)*, *Kozlowski et al. (2013)*, *Cimler et al. (2018)* or non-standard optimization techniques by: *Gavalec & Tomaskova (2010)*, *Bacovsky, Gavalec & Tomaskova (2013)*, *Tomaskova & Gavalec (2013, 2014)*, *Gavalec, Plavka & Tomaskova (2014)*, *Gavalec, Mls & Tomaskova (2015)*, *Cimler et al. (2017)*, *Oudah, Jabeen & Dixon (2018)*.

Some authors have attempted to provide a solution for process model analysis. For example *Melao & Pidd (2000)* discussed the strengths and limitations of the various modelling approaches used in business process transformation. The article by *Glassey (2008)* compares three process modelling processes used in case studies. The article by *Sadiq & Orlowska (2000)* analyze process models using graph reduction techniques. Other authors like *Van der Aalst et al. (2007)*, *Krogstie, Sindre & Jorgensen (2006)* use specific tools, frameworks and methods for process analysis and modelling.

### Business process modelling

Today, process modelling and business process management (BPMN) have a significant impact. Process modelling is currently a mainly graphical representation of processes, e.g. in what order particular activities should be implemented and what inputs and outputs the processes require for proper functioning. The primary goal of process modelling is to increase the efficiency and effectiveness of the entire process as well as partial activities. Many business process modelling techniques have been proposed over the last decades, so the article *Recker et al. (2009)* comparatively assesses representational analyses of 12 popular process modelling techniques to provide insights into the extent to which they differ from each other. The review business process modelling literature and describe the leading process modelling techniques falling to and before 2004 are published in the articl *Aguilar-Saven (2004)*. The topic of visualization of business process model has been investigated in publication *Dani, Freitas & Thom (2019)*, where the authors performed a systematic literature review of the topic "visualization of business process models". *Kalogirou (2003)* is a particularly fascinating article that illustrates how AI techniques might play an essential role in the modelling and prediction of the performance and control of the combustion process. Although BPM initially focused mainly on the

industrial, service and business sectors, it has also appeared in other sectors in recent years. The popularity of BPMN has been confirmed by articles such as *Zarour et al. (2019)*, which presents the current state-of-the-art of BPN extensions. Publication *De Ramon Fernandez, Ruiz Fernandez & Sabuco Garcia (2019)* deals with the optimization of clinical processes.

### Business process model and notation

Business process model and notation is a language for creating business process models *Silver (2009)*. Under the auspices of the Object Management Group (OMG), the Business Process Management Initiative (BPMI) created the BPMN as an open standard in 2004 by the first version 1.0. In 2005, BPMI merged with the Object Management Group (OMG), and the following year, the latter issued the BPMN specification document. In 2010, BPMN version 2.0 was developed, and the current version of BPMN 2.0.2 was released in December 2013. History of BPMN and notation development is a frequent topic of BPMN publications, we can mention *Nisler & Tomaskova (2017)*, *Kocbek et al. (2015)*, *Chinosi & Trombetta (2012)*, *White (2008)*, *Van der Aalst, Adriansyah & Van Dongen (2012)* and *Recker (2012)*. BPMN is similar to flowcharts and is based on the concept of Petri nets, but it is a more sophisticated and user-friendly language. The graphic form of BPMN makes it understandable even for non-experts. In BPMN, we distinguish several types of elements that we can use in modelling. The specific standards link these elements. In the base classification, we define four groups of items. These are Flow Objects, Connecting Objects, Swimlanes and Artifacts, see *The Object Management Group (2011)*.

## Operational Reserach

Operational Research (OR) is the well-known approach of using analytical and advanced methods to help make the best possible decisions. As early as 1980, Article by authors *Shannon, Long & Buckles (1980)* presented the results of a survey of the perception of the usefulness and knowledge of the 12 OR methodologies commonly used in the practice of industrial engineering. The article by *Dubey (2010)* defines the relationship between OR and another branch of sciences. The article *Gu, Goetschalckx & McGinnis (2010)* presents a detailed survey of the research on warehouse design, performance evaluation, practical case studies, and computational support tools. The article *Negahban & Smith (2014)* provided a review of discrete event simulation publications with a particular focus on applications in manufacturing.

OR methods are often associated with new technologies. In article *Sarac, Absi & Dauzère-Pérès (2010)*, a state-of-the-art on RFID technology deployments in supply chains was given to analyze the impact on the supply chain performance. *Xu, Wang & Newman (2011)*, in their article, tries to identify future trends of computer-aided process planning (CAPP). Dynamic ride-share systems is investigated in the article *Agatz et al. (2012)*.

### Linear programming

One of the most popular areas of OR in practice is linear programming (LP). The mathematical model of linear programming tasks contains a single linear purpose

function, and the actual constraints of the problem are described only by linear equations and inequalities. These tasks are most often encountered in economic practices. Linear programming has been described in several books: *Dantzig (1998)*, *Schrijver (1998)*, *Dorfman, Samuelson & Solow (1987)*.

### Multicriterial decision making

The solving of multi-criteria decision-making (MCDM) tasks comprises the search for optimal values of the unknowns, which are simultaneously assessed according to several often contradictory criteria. Thus, the mathematical model of multi-criteria decision problems contains several purpose functions. Depending on how the sets of decision variants are defined, we are talking about the tasks of multi-criteria linear programming or multi-criteria evaluation of options. A review of applications of Analytic Hierarchy Process in operational management is inverstigated in *Subramanian & Ramanathan (2012)*. The article *Velasquez & Hester (2013)* performs a literature review of common Multi-Criteria Decision Making methods. The authors present the results of a bibliometric-based survey on AHP and TOPSIS techniques in publication *Zyoud & Fuchs-Hanusch (2017)*.

### Project planning

Project management tasks consist of several separate activities that are interdependent and may be run simultaneously. The most commonly used method is the so-called network analysis, where a network graph is created from the left chronologically arranged project activities representing the project life cycle. The longest possible path from the beginning to the end of the project is recorded by "the critical path". The non-observance of this path will lead to a slowing down of the whole project, whose time duration is to be optimized. The optimistic, pessimistic, and most probable estimate of the implementation of the entire project is determined. The article *Nutt (1983)* relates the project planning process and implementation. Critical Path Method (CPM) is found in the article *Jaafari (1984)*, to be equally useful as a planning tool for linear or repetitive projects.

The Resource-Constrained Project Scheduling Problem (RCPSP) is a general problem in scheduling. The article *Pellerin, Perrier & Berthaut (2020)* examines the general tendency of moving from pure metaheuristic methods to solving the RCPSP to hybrid methods that rely on different metaheuristic strategies (*Cimr, Cimler & Tomaskova, 2018*).

### Nonlinear programming

Nonlinear programming is the case when the purpose function is not linear. Tasks then often have a large number of local extremes and often also have great difficulty finding them.

### Dynamic programming

If constraints are functions of some parameter, which is most often time, we are talking about dynamic programming. This approach deals with the modelling of more complex multi-stage optimization problems divisible into related sub-problems. Depending on the time parameter, the system is always in one of the acceptable states during the process.

At certain times it is necessary to choose from a set of possible decisions, which again results in the transition to the next state. We call the strategy a sequence of these states of the system and choices, looking for the course with the best valuation. Simulations are often used to model and analyze the operation of complex systems without realization and in less than real-time.

- Queuing theory is a type of dynamic programming task. It deals with streamlining the functioning of systems in which it is necessary to gradually serve all units whose requirements are continuously met on so-called service lines. The challenge is to find the most effective way to handle these requirements.
- Inventory management models address the issue of optimizing the supply process and the volume of inventory stored. Costs associated with ordering, issuing, and keeping stocks in stock should be minimized.

### Stochastic programming

Stochastic programming deals with optimization problems in which they act as parameters of their constraints of random variables. Probabilistic calculus methods solve these problems, and their results have the character of random variables. Stochastic processes can also be ranked among tasks with the input data uncertainties. This approach is used to describe the behavior of systems evolving. We are talking about stochastic processes, a special case is the so-called Markov chains and Markov processes. Basic books on this topic are, for example: *Kall, Wallace & Kall (1994)*, *Birge & Louveaux (2011)*, *Shapiro, Dentcheva & Ruszczyński (2014)*.

## RESEARCH METHODOLOGY

*Kitchenham & Charters (2007)* highlighted three essential elements for a systematic literary review: the determination of the research question(s), the organisation of an unbiased and extensive analysis of related publications, and the determination of precise criteria of inclusion and exclusion.

We identified three research questions:

- Research question 1 (R1): Greater adaptability of BPMN elements causes greater application of this notation in publications.
- Research question 2 (R2): The connection between BPMN and OR methods is most often applied to the business and economics areas.
- Research question 3 (R3): The queue theory is the most widely used method in BPMN processes.

The analysis process and criteria are given in the following relevant subsections.

### Eligibility criteria

This study included publications listed in the Web Of Science (WOS) database of Clarivate Analytics that were published between 1 January 2004 and 18 May 2020. The year 2004 was selected as this is when BPMN was created by BPMI.

**Table 1 Web of Science Core Collection Indexes.**

| Indexes | Abbreviation |
|---|---|
| Science citation index expanded | (SCI-EXPANDED) |
| Social sciences citation index | (SSCI) |
| Arts & humanities citation index | (A&HCI) |
| Conference proceedings citation index—science | (CPCI-S) |
| Conference proceedings citation index—social science & humanities | (CPCI-SSH) |
| Book citation index—science | (BKCI-S) |
| Book citation index—social sciences & humanities | (BKCI-SSH) |
| Emerging sources citation index | (ESCI) |
| Current chemical reactions | (CCR-EXPANDED) |
| Index chemicus | (IC) |

Exclusion criteria (EC) are:

- EC1 = The publication was published in a language other than English.
- EC2 = The full text of the publications was not available.
- EC3 = The publication did not coincide with the topic of systematic research.
- EC4 = BPMN was used only as a presentation tool and not as part of the research.

## Information sources and search

The primary source of information for the study was the database Web Of Science (WOS) of Clarivate Analytics. An advanced search was performed for the search query mentiones below. The search was performed in the Topics (TS) section.

Especially, the CORE database with the indexes listed in Table 1 was selected. The search was performed for 'All document types,' 'All languages' and the years 2004–2020.

## Study selection

The first step of the review process involved title and abstract screening, followed by a full-text review of the remaining articles. Two independent assessors verified the results of the title and abstract screening and the full-text review. One assessed the suitability of the results from the perspective of OR and the other from an IT perspective, i.e. whether it was BPMN notation and its use. Articles were included if they met all the following criteria: (i) they used an OR method, (ii) a BPMN model was used and (iii) the complete text was available in English (abstracts, commentaries, letters and unpublished data were excluded). Studies were not excluded based on their methodological quality.

The selected publications were examined from many perspectives, and each contribution was coded according to different criteria. This study aimed to enhance the discipline's fundamental progress in understanding the link between OR methods and BPMN. The results of this study could encourage scientists to use OR methods for process analysis.

| Table 2 Electronic search strategy in WoS. | |
|---|---|
| **Query** | **Results** |
| TS=(Computer AND programming AND BPMN) | 6 |
| TS=(Decision AND Analysis AND BPMN) | 40 |
| TS=(Decision AND theory AND BPMN) | 7 |
| TS=(Dynamic AND programming AND BPMN) | 4 |
| TS=(Heuristic AND programming AND BPMN) | 0 |
| TS=(Hypothesis AND testing AND BPMN) | 2 |
| TS=(Inventory AND control AND BPMN) | 1 |
| TS=(Linear AND regression AND BPMN) | 2 |
| TS=(Linear AND programming AND BPMN) | 3 |
| TS=(Math AND analysis AND BPMN) | 0 |
| TS=(Math AND programming AND BPMN) | 0 |
| TS=(Network AND analysis AND BPMN) | 23 |
| TS=(Nonlinear AND programming AND BPMN) | 0 |
| TS=(PERT AND BPMN) | 0 |
| TS=(Probability AND BPMN) | 14 |
| TS=(Queuing AND BPMN) | 9 |
| TS=(Statistic AND BPMN) | 2 |
| TS=(Stochastic AND processes AND BPMN) | 15 |

A limitation of this review was restricting the included articles to English-language publications that looked at process analysis using OR and BPMN published between 1 January 2004 and 18 May 2020. Relevant studies in other languages or published after 18 May 2020 may have been omitted.

## Data collection process

Data was collected based on keywords selected from the article *Lane, Mansour & Harpell (1993)*, which analyzed the quantitative techniques of Operation Research. From this document, the 18 Operation Research methods were selected and listed in the Table 2.

The results were further categorized as to whether they corresponded to the given keywords and their meaning. The main results of the systematic literature review were obtained by analyzing by the two main guidelines of PRISMA: *Moher et al. (2009)* and MECIR: *Higgins et al. (2018)*.

## Synthesis of results

The individual studies were subjected to bibliometric analysis and then the studies were assessed according to the content and methods used. The bibliometric analysis describes and analyses up to date research. It aims at summarizing the latest progress in the field by quantitatively investigating the literature. This method provides a vast canvas of knowledge from the micro-level (institutes, researchers, and campuses) to the macro-level (countries and continents) *Mryglod et al. (2013)*. Frequency analysis was used to find the most common scientific areas, the countries with the most publications and the most

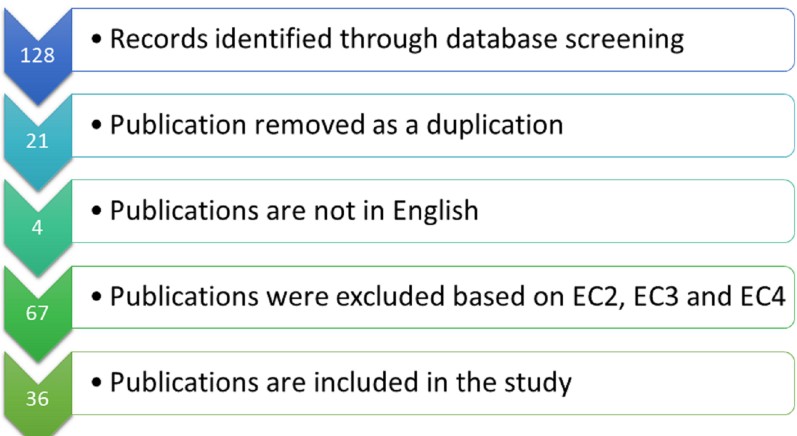

**Figure 1  Overview of the systematic review.**

common keywords. Science mapping was performing using the VOS viewer, Venn diagrams and bar and bubble graphs, *Van Eck et al. (2010)*, *Cobo et al. (2011)*.

The Venn/Euler diagram graphically represents the relationships of the largest set of keywords. Euler diagrams are considered to be an effective means of visualizing containment, intersection, and exclusion. The goal of this type of graph is to communicate scientific results visually. Leonhard Euler first popularized the principle of labeled closed curves in the article *Euler (1775)* Alternative names for Euler diagrams include 'Euler circles.' They can also be incorrectly called Venn diagrams. Venn diagrams require all possible curve intersections to be present, so can be seen as a subset of Euler diagrams, that is, every Venn diagram is a Euler diagram, but not every Euler diagram is a Venn diagram. John Venn introduced Venn diagrams a hundred years after Euler in the article *Venn (1880)*. Venn diagram is a schematic graph used in logic theory to depict collections of sets and represent their relationships.

## RESULTS

The initial search resulted in 128 articles. After removing duplicates, 107 were left that underwent title and abstract screening. After screening, 61 articles remained that underwent full-text review. The final number of included articles for information abstraction was 36. Overview of the number of publications according to exclusion criteria is shown in Fig. 1.

Eighteen keywords selected from the article by *Lane, Mansour & Harpell (1993)* were involved in the study. These keywords have been classified according to whether a publication meeting a study condition has been found for them. Only for 13 keywords were found a publication suitable for this study, as can be seen in Table 2

### Categorization of publications based on the clarivate analytics

Journals and books covered by the Web of Science Core Collection were assigned to at least one Web of Science category. Each Web of Science category was mapped to
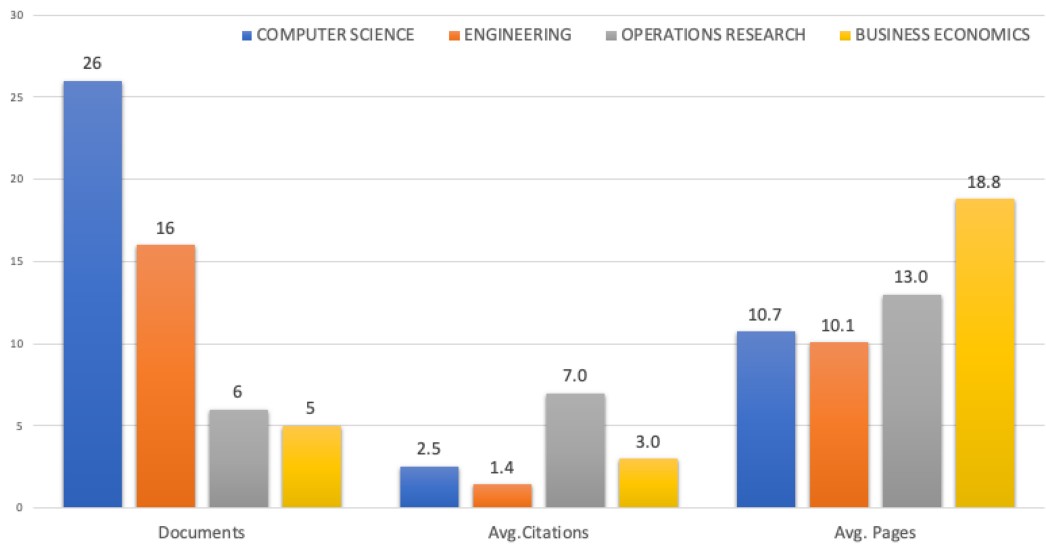

**Figure 2  Research areas of selected publications, documents, average citations and average pages.**

one research area *Clarivate Analytics (2019)*. The research areas for the selected publications were:

- COMPUTER SCIENCE (CS)
- ENGINEERING (En)
- OPERATIONAL RESEARCH MANAGEMENT SCIENCE (OR)
- BUSINESS ECONOMICS (BE)
- ROBOTICS (Ro)
- AUTOMATION CONTROL SYSTEMS (ACS)
- TELECOMMUNICATIONS (Te)
- TRANSPORTATION (Tr)

We selected four main groups, for which we compiled a bar graph and a Venn diagram after analysis. We chose the number of four research areas for representation in the Venn diagram; four sets are still well arranged. Another argument was the number of publications in other areas, where the set "ROBOTICS" contains two documents and the sets 'AUTOMATION CONTROL SYSTEMS,' 'TELECOMMUNICATIONS' and 'TRANSPORTATION' each one document.

Bar graph on Fig. 2 is based on frequency analysis and contains the total number of publications in a given research area, their average number of citations, and the corresponding average number of pages per article. The graph shows the results by type of purpose. The first part shows the frequency of documents for each research areas. The second part focuses on the average number of citations, and the third shows the average number of pages per article.

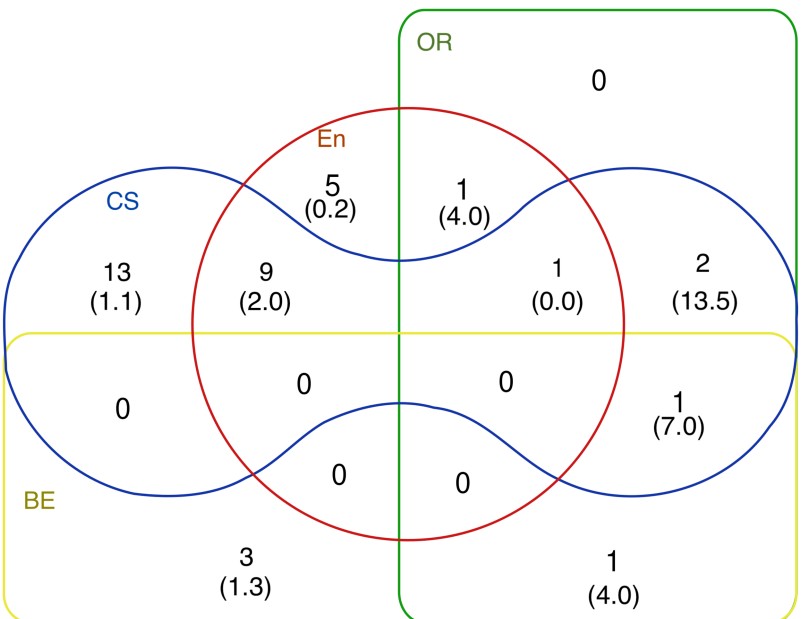

**Figure 3 Venn diagram of research areas of selected publications, the average number of citations is listed in the parentheses.**

The Venn diagram, in Fig. 3, shows selected four research areas as sets, including their intersection areas. In a specific area, we also stated the relevant number of documents and their average number of citations.

This part of the bibliometric analysis showed us the answer to the research question R2. Although BPMN was explicitly designed for corporate analysis and economic analysis, and Operational Research focuses primarily on addressing managerial decisions, most publications were not in the field of business economics (BE). Surprisingly, this area actually has the fewest publications. The field of computer science had the most papers, and papers in the field of OR had the most citations. The field of BE had the most extended publications, however, i.e. the average number of pages per paper.

Result1: Research question R2—not confirmed.

## Year of publication

Figure 4 illustrates the distribution over time of the selected publications with BPMN milestones. The BPMN versions adoption dates, taken from *OMG.org (2018)*, complements this figure.

The different BPMN versions brought more or fewer changes in notation. While the changes between BPMN 1.0 and BPMN 1.2 were rather consmetics, e.g. renaming 'Rule' elements to 'Conditional' or slight increasing the number of elements from 48 to 55. The arrival of BPMN 2.0 was a major breakthrough and represented the largest revision of BPMN since its inception. In this version, it is possible to create a new 'Choreography model,' 'Collaborations model' and 'Conversation model' in BPMN in addition to collaborative processes and internal (private) business processes. Events are now divided into 'interrupted' and 'non-interrupted' and 'catching' and 'throwing.' The message type is

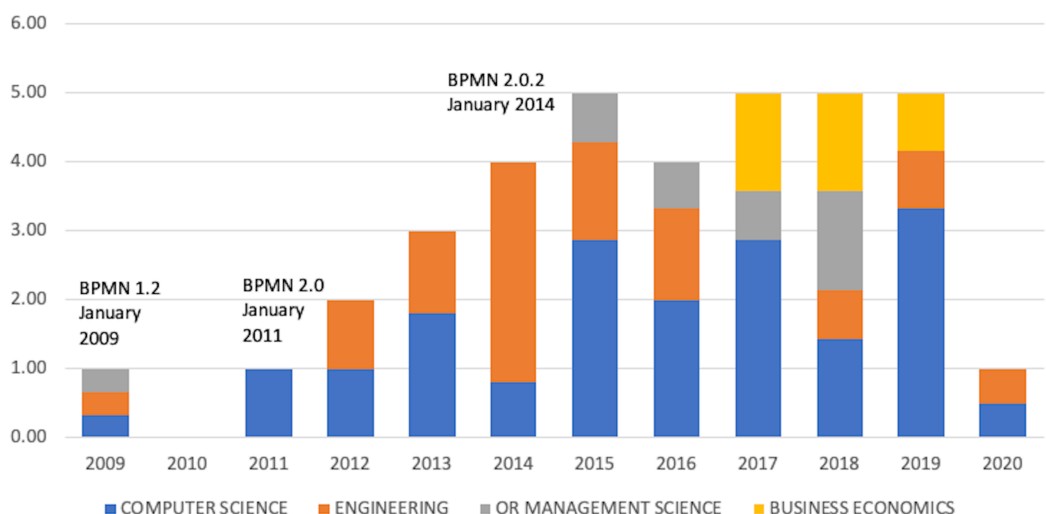

**Figure 4** **Distribution of the publications by year and their representation in research areas.**

newly introduced, and the data object has three specifications. BPMN 2.0 contains 116 elements. BPMN 2.0.2 included only minor modifications in terms of typos.

Given the magnitude of changes between the different versions of the BPMN notation, the sharp increase in publications following the introduction of the BPMN 2.0 notation can be considered a confirmation of research question R1. It is very interesting that publications in the field of BE did not appear until 2017.

Result: Research question R1—confirmed.

The average number of citations of the analysed documents was 2.22. The first quartile was 0, and the third quartile was 3.75. The median was equal to 1 and data variability above the third quartile was limited to seven citations. We identified two outliers values: 12 citations for *Hasic, De Smedt & Vanthienen (2018)* and 15 citations for article *Wu et al. (2015)*.

## Author analyses

Bibliometric analysis cannot be done without review by the authors. We focused on illustrating co-authorship. The total number of authors of publications selected for this study was 84: al achhab, m (1), aouina, zk (1), ayani, r (1), aysolmaz, b (1), bahaweres, rb (1), batoulis, k (1), ben ayed, ne (1), ben said, l (1), ben-abdallah, h (3), bisogno, s (1), bocciarelli, p (1), boukadi, k (1), braghetto, kr (1), burattin, a (1), calabrese, a (1), ceballos, hg (2), chien, cf (1), cho, sy (1), creese, s (1), cunha, p (1), d'ambrogio, a (1), d'ambrogio, sa (1), de lara, j (1), de smedt, j (2), demirors, o (1), duran, f (2), el hichami, o (1), el mohajir, b (1), ferreira, je (1), figl, k (1), fitriyah, a (1), flores-solorio, v (2), fookes, c (1), garcia-vazquez, jp (1), ghiron, nl (1), ghlala, r (1), gomez-martinez, e (1), hansen, z (1), hansen, znl (3), happa, j (1), hasic, f (2), herbert, lt (8), holm, g (1), iren, d (1), jacobsen, p (3), jobczyk, k (1), kamrani, f (1), khlif, w (2), kluza, k (1), ligeza, a (1), manuel vara, j (1), marcos, e (1), mazhar, s (1), mendling, j (1), mendoza morales, le (1), mengersen, k (1), monsalve, c (1), moradi, f (1), naoum, m (1), onggo, bss (1), pablo garcia, j (1),

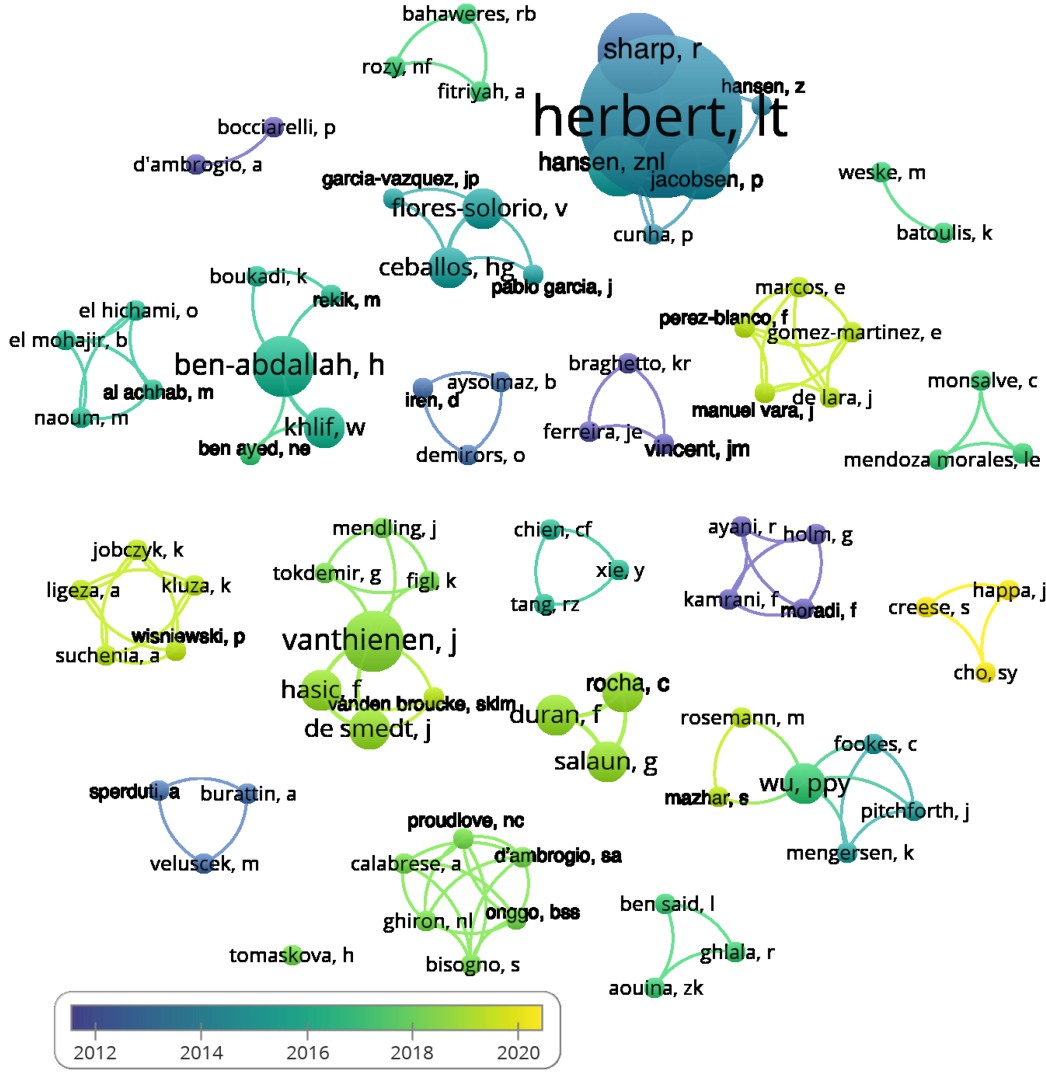

**Figure 5 Division of authors into groups according to co-authorship.**

perez-blanco, f (1), pitchforth, j (1), proudlove, nc (1), rekik, m (1), rocha, c (2), rosemann, m (1), rozy, nf (1), salaun, g (2), sharp, r (4), sperduti, a (1), suchenia, a (1), tang, rz (1), tokdemir, g (1), tomaskova, h (1), vanden broucke, sklm (1), vanthienen, j (3), veluscek, m (1), villavicencio, m (1), vincent, jm (1), weske, m (1), wisniewski, p (1), wu, ppy (2), xie, y (1).

These authors formed different sized groups, as can be seen in Fig. 5. We grouped the authors according to their co-authors' collaborations with a curve connecting the co-authors. The size of the node of this connection corresponds to the number of documents by the given author. The colours used to distinguish the authors were created using the average years of the publication of their papers.

For the authors' average publication years, the first quartile was 2015, the third quartile was 2018.5 and the median was 2017. The variability outside the lower and upper quartiles

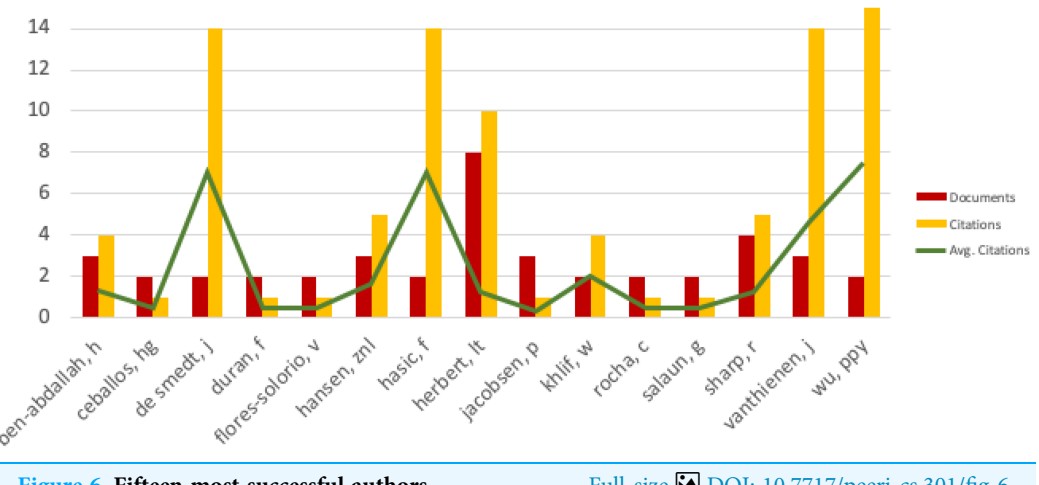

**Figure 6 Fifteen most successful authors.** 

was given by 2011 and 2020. We identified one outlier value corresponding to the year 2009.

The most prominent groups were around the authors listed in Fig. 6. This figure also contains the number of documents by the authors, their total number of citations and their average value.

According to this analysis Wu, P. Y. had the highest number of citations (7.5), followed by De Smedt, J. (7) and Hasic, F. (7). Herbert, L.T. had the most documents (8) and Tomaskova, H. had no co-author connections.

The authors were also analyzed in terms of their country or region affiliations. A total of 25 countries were identified and their location, including the number of relevant publications, are shown in Fig. 7. The countries with the highest number of affiliated publications were Denmark (8) and Tunisia (4), followed by Belgium, France, Saudi Arabia, Italy and Spain, who all had three.

## Keywords analysis

The keywords were categorized according to those identified by the published authors and the keywords PLUS assigned by Clarivate Analytics databases. The data in KeyWords Plus are words or phrases that frequently appear in the titles of an article's references but do not appear in the title of the item itself. Based upon a special algorithm that is unique to Clarivate Analytics databases, KeyWords Plus enhances the power of cited-reference searching by searching across disciplines for all the articles that have cited references in common, more information is on the web link *Clarivate Analytics (2018)*. A total of 130 unique keywords and 46 unique KeyWords Plus keywords were found for selected publications.

A total of 130 author keywords were mentioned in the publications and a general view of their interconnection can be seen in Fig. 8.

Below is a list of all author keywords with the number of the weight-link to other keywords: activity theory (4), affiliation (6), agent based model (4), agent-based systems engineering (3), airport passenger facilitation (8), atl (5), automated verification (4),

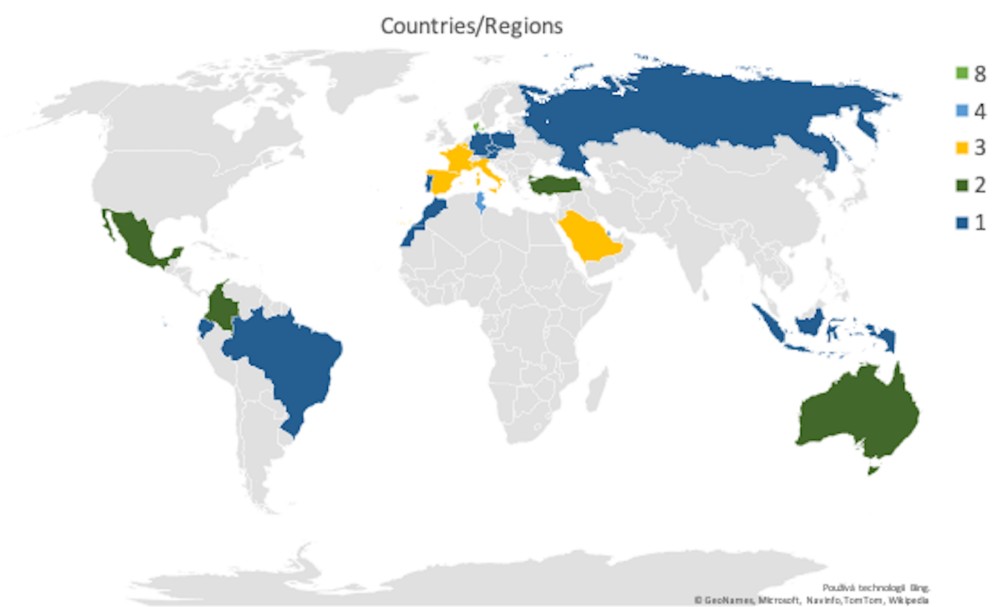

**Figure 7  Location of publications in the world, own processing.**

bayesian network (4), bayesian networks (4), bpm (6), bpmn (60), bpmn business processes (4), bpmn extension (3), bpmn model restructuring (5), business process (18), business process automation (3), business process management (13), business process model (5), business process model measures (3), business process modelling notation (4), business process optimisation (5), business process outsourcing (3), business processes (3), cloud computing (3), clustering (5), communication theory (11), configurable reference model (8), consequence modelling and management (10), contextual factors (8), cycle time (4), decision making (15), decision mining (3), decision model and notation (3), decision modeling (4), decision modelling (5), dikw (11), discrete-event simulation (4), dmn (15), effort prediction model (3), engineering agent-based systems (3), engineering systems (6), enterprise risk management (4), eqn (5), evolutionary algorithm (2), evolutionary algorithms (5), facilitated modelling (4), fault tree analysis (6), fault tree generation (6), flow (8), formal risk analysis (6), genetic algorithm (3), healthcare (4), hierarchical clustering (6), incident response (11), integrated modelling (5), interviews (11), jeqn (5), knowledge discovery (6), knowledge management (11), knowledge rediscovery (6), licenses (11), maude (7), mc-dmn (5), mcdm (5), mda (5), metrics (5), model checking (4), model transformations (5), model-driven architecture (4), model-driven engineering (5), modelling (4), object modeling (4), optimisation (6), organizational mining (3), performance (5), performance evaluation (3), petri nets (5), pproduction optimisation (2), preference to criteria (5), prism (8), probability (2), process configuration (8), process enhancement (3), process chain network (5), process merging (8), process mining (6), process modeling (4), process modelling (5), project management (3), qualitative analysis (4), quantitative model checking (10), quantitative service analysis (6),

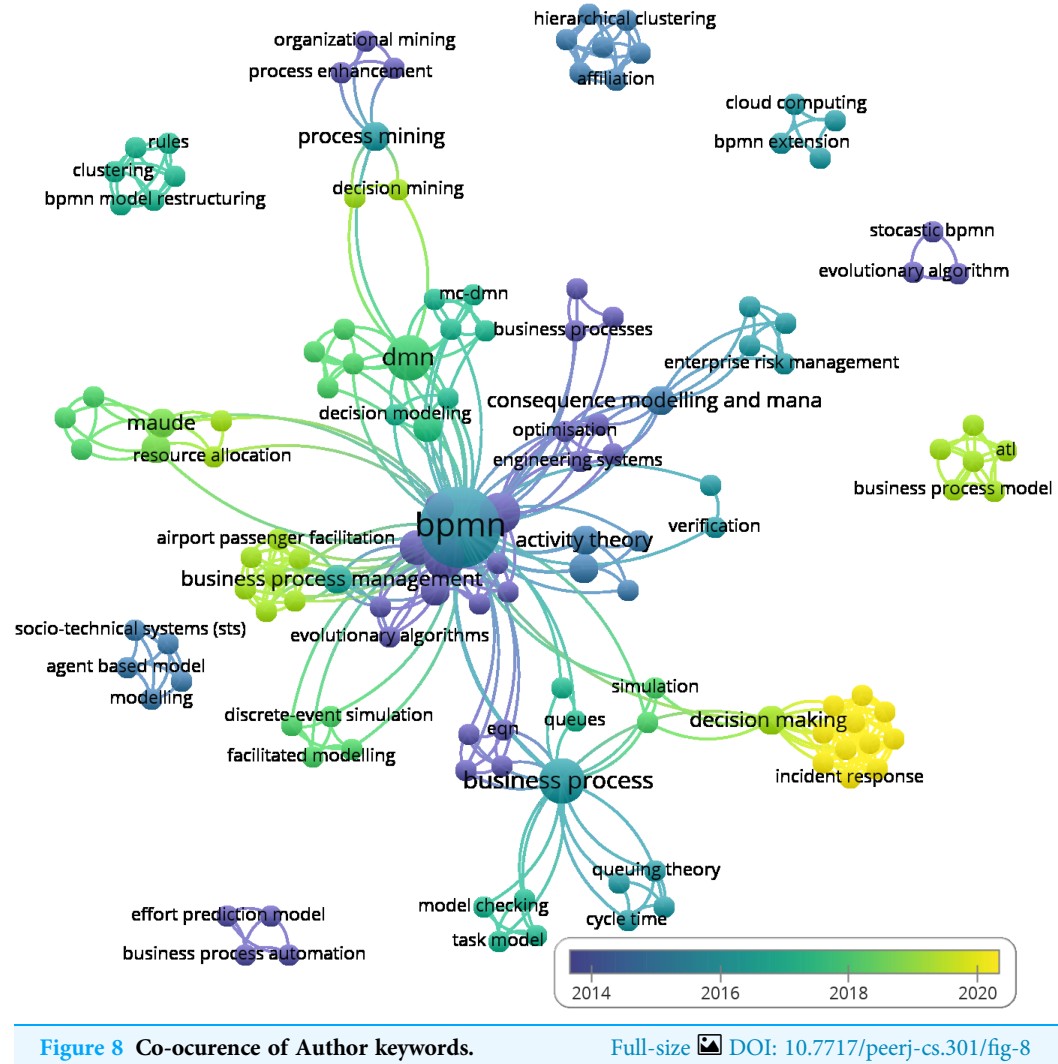

**Figure 8 Co-ocurence of Author keywords.**

quantitative workflow analysis (4), queues (3), queuing theory (4), reliability analysis and risk assessment methods (4), resource allocation (4), restructuring (6), rewriting logic (7), rules (5), safety assessment software tools (4), safety management and decision making (4), security (11), security operation center (11), sense-making (11), separation of concerns (5), service engineering (6), scheduling (4), simulation (4), simulations (3), social network (5), social network analysis (3), social network model (6), socio-technical systems (sts) (4), soundness (4), space-sensitive process model (8), statistical model checking (4), stocastic bpmn (2), stochastic automata network (3), stochastic bpmn (11), stochastic model checking (13), stochastic modeling and analysis (4), structural and semantic aspects (5), tacit knowledge (11), task analysis (11), task assignment (4), task model (4), timed automata (4), topsis (5), verification (2).

As you can see in the figure, most of the author's keywords are directly or indirectly linked with the term 'BPMN', but there are also isolated groups. In the following text,

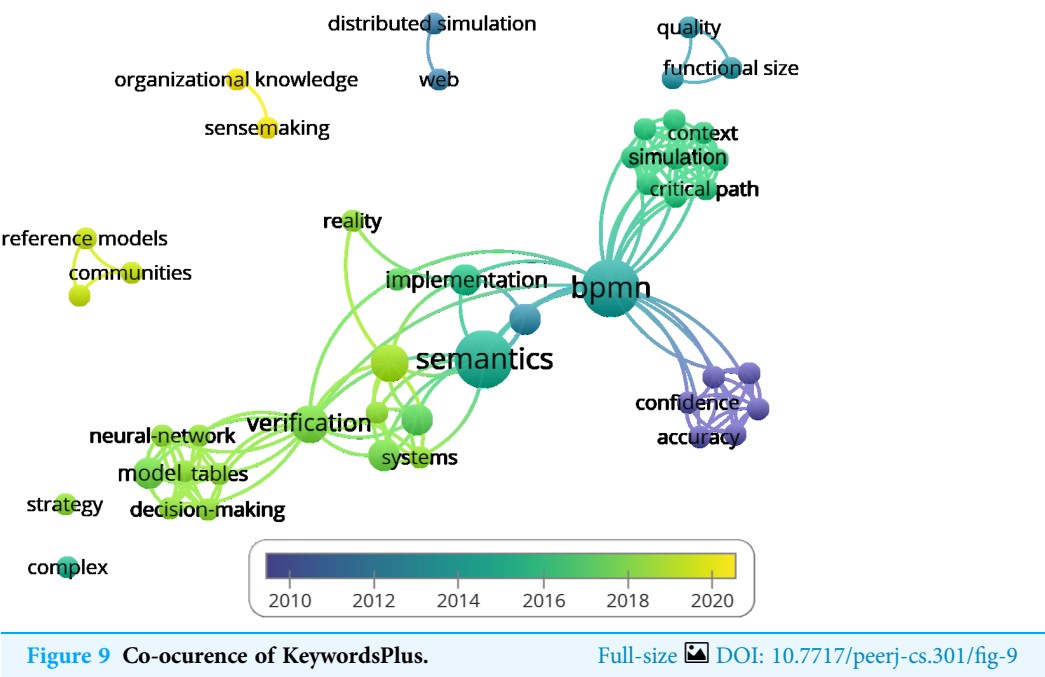

**Figure 9 Co-ocurence of KeywordsPlus.**

we've listed separate keyword groups. We've added a year of publication, a number of citations, and a specific document to which the keywords belong.

- 2013; citations; (business process automation; business process model measures; effort prediction model; project management) *Aysolmaz, Iren & Demirors (2013).*
- 2014; citation; (evolutionary algorithm; pproduction optimisation; stocastic bpmn) *Herbert et al. (2014),*
- 2015; citations; (agent based model; bayesian network; business process modelling notation; modelling; socio-technical systems (sts)) *Wu et al. (2015),*
- 2015; citation; (affiliation; bpm; hierarchical clustering; knowledge discovery; knowledge rediscovery; restructuring; social network model) *Khlif & Ben-Abdallah (2015),*
- 2016; citations; (bpmn extension; business process outsourcing; cloud computing; genetic algorithm) *Rekik, Boukadi & Ben-Abdallah (2016).*
- 2017; citations; (bpmn model restructuring; clustering; metrics; rules; social network; structural and semantic aspects) *Khlif, Ben-Abdallah & Ben Ayed (2017).*
- 2019; citations; (atl; business process model; model transformations; model-driven engineering; petri nets; process chain network) *Gómez-Martnez et al. (2019).*

As mentioned above, there were only 46 KeyWords Plus keywords (the number of links to other keywords is given in parentheses after the keyword): accuracy (6), ambiguity (6), automation (3), bpmn (20), business process models (6), checking (6), cognitive effectiveness (7), communities (2), complex (0), confidence (6), context (9), critical path (9), decision-making (7), design (7), dimensions (7), distributed simulation (1), framework (8), functional size (2), group creativity (6), identification (9),

implementation (5), information (6), integration (2), model (7), neural-network (7), organizational knowledge (1), patterns (6), performance (9), process execution (9), process models (9), productivity (2), quality (2), reality (2), reference models (2), resources (9), risk (6), science research (2), semantics (9), sensemaking (1), simulation (9), strategy (0), systems (6), tables (7), verification (15), web (1), workflow (9).

As can be seen in Fig. 9, these keywords are far more separate from each other compared to the author's keywords.

## Classification of articles by methodology

Based on the expert assessment, we examined the documents regarding the methods and approaches used. We created seven groups corresponding to a method or approach that was an essential part of the publication: probabilistic models, Decision Model and Notation (DMN), dynamic task assignment problem, evolutionary and genetics algorithms, queuing theory, social networks and others. These groups were also based on keyword analysis, as some separate groups of copyright keywords belong to highly unique articles. We assigned each document to just one group. That is in contradiction to research areas, where one article can be attributed to more than one research area. The individual documents and their division between research areas and methodological groups can be seen in Table 3. We further analyzed the documents regarding their years of publication and plotted a bubble graph (Fig. 10) with the publication years on the $x$.axis and the methodological groups on the $y$-axis. The appropriate number of publications corresponding to the given year and the group is indicated in the respective bubble. This quantity is also graphically represented by the size of the given bubble.

The largest group consisted of 10 publications on DMN and BPMN. Given the initiate year of DMN, this is the most significant approach serving with BPMN. DMN 1.0 was made available to the public in September 2015, the OMG group released DMN 1.1 in June 2016, DMN 1.2 was released in January 2019 and the latest version of DMN 1.3 was released in March 2020. The latest version did not affect this systematic search; however, the growth of publications since 2017 (see Fig. 10, for example, was undoubtedly be affected by the DMN update.

We only assigned four documents to the methodological group focused on queue theory (See Table 3 and Fig. 10). The specific articles are listed in the following section under the appropriate heading. As the largest group was the DMN and BPMN group, we can thus rule out research question R3.

Result: Research question R3—not confirmed.

The methods, techniques and approaches used in the included publications are listed in the following section.

### Probabilistic models

The probabilistic model can be used to make decisions when the activity reaches an exclusive splitting gateway and the activity's subject must decide between alternative

**Table 3 Documents division according to the research areas and methodological groups.**

| Indexes | | | Abbreviation |
|---|---|---|---|
| Science Citation Index Expanded | | | (SCI-EXPANDED) |
| Social Sciences Citation Index | | | (SSCI) |

| | Computer science (26) | Engineering (16) | Operational Research (6) | Business economics (5) |
|---|---|---|---|---|
| Probabilistic models (5) | *Herbert & Sharp (2012, 2013), Ceballos, Flores-Solorio & Garcia-Vazquez (2015), Ceballos, Flores-Solorio & Pablo Garcia (2015), Naoum et al. (2016)* | *Herbert & Sharp (2012), Herbert & Sharp (2013), Ceballos, Flores-Solorio & Garcia-Vazquez (2015)* | | |
| DMN (10) | *Batoulis & Weske (2017), Ghlala, Aouina & Ben Said (2017), Hasic, De Smedt & Vanthienen (2018), De Smedt et al. (2019), Durán, Rocha & Salaün (2019), Suchenia et al. (2019), Cho, Happa & Creese (2020)* | *Figl et al. (2018), Suchenia et al. (2019), Cho, Happa & Creese (2020)* | *Batoulis & Weske (2017), Hasic, De Smedt & Vanthienen (2018* | *Batoulis & Weske (2017), Tomaskova (2018), Mazhar, Wu & Rosemann (2018)* |
| Dynamic task assignment problem (1) | *Xie, Chien & Tang (2016)* | *Xie, Chien & Tang (2016)* | | |
| Evolutionary and genetic algorithms (5) | *Herbert & Sharp (2014b), Rekik, Boukadi & Ben-Abdallah (2016)* | *Herbert & Sharp (2014b), Herbert et al. (2014), Herbert, Hansen & Jacobsen (2015), Herbert & Hansen (2016)* | *Herbert & Hansen (2016)* | |
| Queuing theory (4) | *Bocciarelli & D'Ambrogio (2012), Bahaweres, Fitriyah & Rozy (2017), Gómez-Martnez et al. (2019)* | *Bocciarelli & D'Ambrogio (2012)* | *Onggo et al. (2018)* | *Onggo et al. (2018)* |
| Social network (2) | *Khlif & Ben-Abdallah (2015)* | | | *Khlif, Ben-Abdallah & Ben Ayed (2017)* |
| Other (9) | *Kamrani et al. (2009), Braghetto, Ferreira & Vincent (2011), Aysolmaz, Iren & Demirors (2013), Burattin, Sperduti & Veluscek (2013), Wu et al. (2015), Mendoza Morales, Monsalve & Villavicencio (2017), Duran, Rocha & Salaun (2018)* | *Kamrani et al. (2009), Burattin, Sperduti & Veluscek (2013), Herbert & Sharp (2014a), Herbert, Hansen & Jacobsen (2014)* | *Kamrani et al. (2009), Wu et al. (2015)* | |

actions. They can be used for predicting or deciding between alternative works based on desirable outcomes. Probabilistic models were presented in the following publications:

- *Herbert & Sharp (2012)*: Quantitative analysis of probabilistic BPMN workflows;
- *Herbert & Sharp (2013)*: Precise quantitative analysis of probabilistic business process model and notation workflows;
- *Ceballos, Flores-Solorio & Garcia-Vazquez (2015)*: Towards Probabilistic Decision Making on Human Activities modeled with Business Process Diagrams;
- *Ceballos, Flores-Solorio & Pablo Garcia (2015)*: A Probabilistic BPMN Normal Form to Model and Advise Human Activities;
- *Naoum et al. (2016)*: A probabilistic method for business process verification: Reachability, Liveness and deadlock detection,

there the (Causal) Bayesian Network or Markov Decision processes were used.

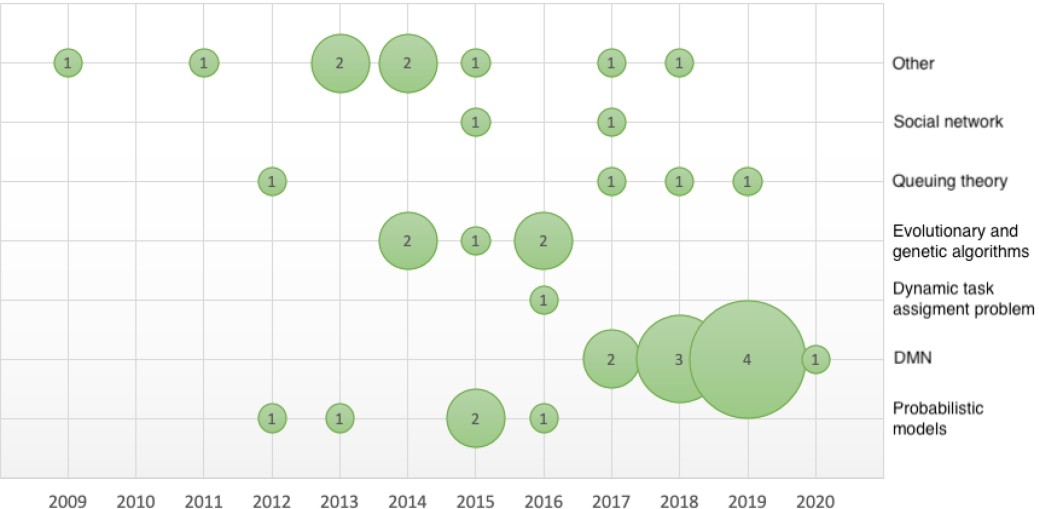

**Figure 10 Buble graph—quantity of article according to methodological groups and publication year.**

### DMN and decision analysis

Decision Model and Notation (DMN) is an industry standard for modeling and executing decisions that are determined by business rules. The association of DMN and BPMN is now common practice:

- *Batoulis & Weske (2017)*: Soundness of decision-aware business processes,
- *De Smedt et al. (2019)*: Holistic discovery of decision models from process execution data,
- *Durán, Rocha & Salaün (2019)*: A rewriting logic approach to resource allocation analysis in business process models,
- *Figl et al. (2018)*: What we know and what we do not know about DMN,
- *Ghlala, Aouina & Ben Said (2017)*: MC-DMN: Meeting MCDM with DMN Involving Multi-criteria Decision-Making in Business Process
- *Hasic, De Smedt & Vanthienen (2018)*: Augmenting processes with decision intelligence: Principles for integrated modelling
- *Cho, Happa & Creese (2020)*: Capturing Tacit Knowledge in Security Operation Centers,
- *Mazhar, Wu & Rosemann (2018)*: Designing complex socio-technical process systems - the airport example,
- *Suchenia et al. (2019)*: Towards knowledge interoperability between the UML, DMN, BPMN and CMMN models
- *Tomaskova (2018)*: Modeling Business Processes for Decision-Making.

Both standards fall under OMG.

### Dynamic task assignment approach

The study : A dynamic task assignment approach based on individual worklists for minimizing the cycle time of business processes by *Xie, Chien & Tang (2016)* develop a

dynamic task assignment approach for minimizing the cycle time of business processes. The contribution of this article lies in developing a dynamic task assignment approach based on queuing theory, individual worklist model, and stochastic theory.

### Evolutionary and genetic algorithms

The evolutionary algorithm was applied in the following publications:

- *Herbert & Sharp (2014b)*: Optimisation of BPMN business models via model checking;
- *Herbert et al. (2014)*: Evolutionary optimization of production materials workflow processes;
- *Herbert, Hansen & Jacobsen (2015)*: Using quantitative stochastic model checking tool to increase safety and improve efficiency in production processes;
- *Herbert & Hansen (2016)*: Restructuring of workflows to minimise errors via stochastic model checking: An automated evolutionary approach;

to optimize the BP diagram, thus looking for a more efficient process. Especially the publication: Specifying business process outsourcing requirements, *Rekik, Boukadi & Ben-Abdallah (2016)*, presented a genetic algorithm to identify most appropriate activities of a business process that should be outsourced.

### Queuing theory

In the article: Comparative analysis of business process litigation using queue theory and simulation (case study: Religious courts of South Jakarta) *Bahaweres, Fitriyah & Rozy (2017)*, *Onggo et al. (2018)*. A BPMN extension to support discrete-event simulation for healthcare applications: an explicit representation of queues, attributes and data-driven decision points *Onggo et al. (2018)* and *Gómez-Martnez et al. (2019)*. Formal Support of Process Chain Networks using Model-driven Engineering and Petri nets *Gómez-Martnez et al. (2019)*, the authors use queuing theory and simulation to compare processes modeled in BPMN. In the article: Automated performance analysis of business processes *Bocciarelli & D'Ambrogio (2012)*, authors presented a BP performance model of EQN (Extended Queueing Network) type.

### Social network

The publications below focus on the application of social network analysis metrics (SNA) to studies of biological interaction networks in informatics.

- *Khlif & Ben-Abdallah (2015)*: Semantic and structural performer clustering in BPMN models transformed into social network models;
- *Khlif, Ben-Abdallah & Ben Ayed (2017)*: A methodology for the semantic and structural restructuring of BPMN models.

### Other approaches

The following publications were unique in their approaches. We can mention for example: Workflow fault tree generation through model checking by *Herbert & Sharp (2014a)* with

FMEA analysis. An effort prediction model based on BPM measures for process by *Aysolmaz, Iren & Demirors (2013)* with Linear multiple regression analysis. Performance evaluation of business processes through a formal transformation to SAN by *Braghetto, Ferreira & Vincent (2011)* using Stochastic Automata Network. Estimating performance of a business process model by *Kamrani et al. (2009)* using a Task assignment approach. Formal verification of business processes as timed automata by *Mendoza Morales, Monsalve & Villavicencio (2017)* convert BPMN to Timed Automata and then perform standard Queuing analysis. Business models enhancement through discovery of roles by *Burattin, Sperduti & Veluscek (2013)*, there the authors have extended the process model to roles, specifically designed role-sharing algorithm. Stochastic analysis of BPMN with time in rewriting logic by *Duran, Rocha & Salaun (2018)* presents a rewriting logic executable specification of BPMN with time and extended with probabilities. SBAT: A STOCHASTIC BPMN ANALYSIS TOOL by *Herbert, Hansen & Jacobsen (2014)* presents SBAT, a tool framework for the modelling and analysis of complex business workflows and A framework for model integration and holistic modelling of socio-technical systems by *Wu et al. (2015)* presents a layered framework for the purposes of integrating different socio-technical systems (STS) models and perspectives into a whole-of-systems model.

## DISCUSSION

We have identified several gaps in the research and issues that need to be addressed in future research. The main gaps concern the research area of business economics. We assumed that this area would be the main and most frequent for the combination of BPMN and OR methods. However, we found that this area could be affected by the absence of specific notation. The relevant publications were written only after the release of version DMN 1.1. The effect of DMN notation will be addressed in future research.

An unexpected gap was a solution to finance and human resources management through OR. We would like to introduce publications *Savku & Weber (2018)* and *Graczyk-Kucharska et al. (2020)* as the pioneering works. The first article added the problem of optimal consumption problem from cash flow with delay and regimes. The authors developed the general analytic model setting and methods for the solution by studying a stochastic optimal control problem using the tools of the maximum principle. They proved the necessary and sufficient maximum principles for a delayed jump-diffusion with regimes under full and partial information. The second publication focused on transversal competencies, which are sets of knowledge, skills and attitudes required for different positions and in different professions. The authors used the method of multivariate additive regression spline together with artificial neural networks to create a model describing the influence of various variables on the acceleration of the acquisition of transverse competencies.

We assume that future research will be influenced by simulation and prediction methods. This study showed the use of Agent-based modelling methods and discrete-event simulations, or probabilistic models and social networks, but neural networks or artificial intelligence methods appeared in any publication. Based on this study, we further expect the use of more sophisticated approaches and the effect of new techniques.

At the same time, it is possible to extend process modelling to inaccurate data using Fuzzy methods.

## CONCLUSION

This paper presented a systematic overview of publications using BPMN and OR methods in process analysis. We analyzed 108 articles, that were selected using the appropriate strings in the advanced search option of in the WOS database. The papers that met the conditions of the study were subjected to various analyzes and were briefly described.

The review showed that the processes modelled by BPMN can be extended or analyzed as probabilistic processes, queue theory, or role and task assignments. Alternatively, processes can be optimized using evolutionary or genetic algorithms. The research also highlighted the need to identify keywords in publications correctly. For example, less than two-thirds of the selected articles contained the keyword BPMN, even though all the documents used this notation. Most of the articles were so-called one-off publications. Only a small number of author teams developed their topic in further continuing publications. Due to this, the average number of citations is relatively low. Due to the average number of citations to the total number of publications in all research areas, documents falling into the field of Operational Research are outstanding; there is an average of seven citations per article.

We analyzed the publications by research area and found that there is great potential for the research area of business economics (BE). Only a few papers were associated with this area (five in total) but all of them had a higher than average number of citations. The first document we included in this research area was published in 2017, that is only in the last quarter of the examined publication years. This focus on BE may have been initiated by the introduction of DMN notation.

Among the authors, smaller collaborating groups around the world were been identified. That groups co-work within the framework of co-authorship and co-citations. We only identified one single-author publication.

The analysis of keywords showed a significant difference between the keywords assigned by the authors and the so-called KeyWords Plus keywords. While the former were almost completely connected across publications, the latter were significantly diversified.

We have pointed out that the introduction of BPMN 2.0 led to an increase in publications using this notation.

## ACKNOWLEDGEMENTS

The authors thank the student M. Kopecký for support in the field of BPMN modeling.

### Funding

The research has been supported by a GACR 18-01246S and by the Faculty of Informatics and Management UHK Specific Research Project. The funders had no role in study design, data collection and analysis, decision to publish, or preparation of the manuscript.

## Grant Disclosures

The following grant information was disclosed by the authors:
GACR 18-01246S.
Faculty of Informatics and Management UHK Specific Research Project.

## Competing Interests

The author declares that they have no competing interests.

## Author Contributions

- Hana Tomaskova conceived and designed the experiments, performed the experiments, analyzed the data, performed the computation work, prepared figures and/or tables, authored or reviewed drafts of the paper, and approved the final draft.
- Gerhard-Wilhelm Weber analyzed the data, authored or reviewed drafts of the paper, and approved the final draft.

## Data Availability

This article does not contain data or code because it is a literature review.

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
