# Peer review of "Approaches combining methods of Operational Research with Business Process Model and Notation: A systematic review"

_PeerJ Computer Science, doi:10.7717/peerj-cs.301_

## Round 0.1 · original submission · Major Revisions

I felt that your manuscript requires some revision before it can be reconsidered and hope that you will submit a revised version of your manuscript that takes into account the comments of the referee(s), which are included at the bottom of this letter.

Reviewer 1 ·

Basic reporting

>> Paper title and introduction should be reviewed in relation to “decision making” focus. This focus is not sustained in the other sections of the work.
>> The Introduction does not make it clear that the article is a systematic review and does not discuss the results obtained either.
>> The Introduction lacks a final paragraph describing the structure of the paper.
>> The paper lacks both a Related Work and a Background Section. Regarding Related Work, none of the other systematic reviews on process modeling are cited. Background Section would be useful since many topics like BPMN, Linear Programming, Multicriteria decision making and so on are incorrectly placed in a section named “Methods”.
>> Regarding BPMN, it is not correct do say that “The OMG create this notation…”. This mistake is repeated in page 2 and 3. Furthermore, it is contradictory with other parts of the paper that properly give credit to BPMI.
>> Figure 3 is called on page 5, but it is only presented on page 9. Thus, the figure and the text are very distant from each other.
>> Tables are not ordered. It skips from Table 2 to Table 5 and then back to Tables 3 and 4.

Experimental design

>> The analysis of the distribution of the publications by year is very interesting and the use of BPMN versions as important milestones was important.
>> The analysis of authors from selected publications are also important, because it helps to discover the community involved with this topic. However, although the table with the results is presented, the authors did not dedicate themselves to analyze these results in detail.
>> In fact, the lack of depth in the analysis is not a characteristic of the authors topic alone. In general, there is a lack of deepening in the discussions of the results found. Therefore, very narrow conclusions were obtained.
>> I do not understand how publications on biological interaction networks (2 papers of Silva and Saraiva) could be included in this systematic review of operational research and BPMN. Considering publications title, it does not look like they are aligned with the systematic review scope.

Validity of the findings

>> “Limitations” does not look like a Section with only 3 lines.
>> The Conclusion does not help to make the contribution of the paper clear. The main findings are poor discussed.
>> The Conclusion lacks a discussion on the opportunity for future work. Therefore, does not identify unresolved questions / gaps / future directions.

Additional comments

Minor comments:
>> The phrase is incomplete in the first paragraph of the Introduction: “Becker et al. (2010)…”
>> The author should review the idea of BPMN as a “graphical writings” (last paragraph of Introduction). It should be more precisely to say that it is a “language”.

Reviewer 2 ·

Basic reporting

The paper provides a systematic literature review of used operational research techniques in Business Process Model and Notation (BPMN) models.

Overall, the paper is well-written and can be considered for publication. I do, however have a few minor comments that need to be addressed before publication:

1) Please try to make Figure 1 more clear, since the ‘records screened’ and ‘records excluded’ activities fall in between the ‘eligibility’ and ‘screening’ phase. Are these not just screening steps? Or did you put those steps on purpose in between those two phases. If so, please explain in the text why.

Experimental design

2) Please explain the criteria for exclusion more clearly, since there are two exclusion activities: ‘records excluded’ and ‘full text articles excluded’. Explain which criteria were used in each of those in the text. It is very important to report this clearly, as the paper is a literature review paper.

3) The papers that include DMN are rather general and do not really belong to field of operational research. There are, however, papers from the time period that you consider that are more quantitative and more closely related to the topic of the paper. I give the references below, I would like to see them included in the DMN section, or in the “other approaches” section, or in a separate discussion section, as the ones that are included now are not really relevant.

a. De Smedt, J., Hasić, F., vanden Broucke, S. K., & Vanthienen, J. (2019). Holistic discovery of decision models from process execution data. Knowledge-Based Systems, 183, 104866.

b. Ghlala, R., Aouina, Z. K., & Said, L. B. (2017). MC-DMN: Meeting MCDM with DMN involving multi-criteria decision-making in business process. In International Conference on Computational Science and Its Applications (pp. 3-16). Springer, Cham.

c. Hasić, F., De Smedt, J., & Vanthienen, J. (2018). Augmenting processes with decision intelligence: Principles for integrated modelling. Decision Support Systems, 107, 1-12.

d. Hasić, F., De Smedt, J., Broucke, S. V., & Asensio, E. S. (2019). Decision as a Service (DaaS): A Service-Oriented Architecture Approach for Decisions in Processes. IEEE Transactions on Services Computing.

Validity of the findings

4) While the reporting of the results of the literature review is done relatively well, I miss a small discussion section about the implications of the results. What broad conclusions can be drawn from the study? Are there any research gaps / opportunities identified? Some broader findings should be included in a small section before the current limitation section.

---

## Round 0.2 · Minor Revisions

Although the reviewers think the paper improved a lot, they still ask for a deeper discussion of future opportunities and related work.

Reviewer 1 ·

Basic reporting

>> Related Work Section needs to be deeper developed. Is there any other kind of systematic review on this topic? Why are the authors talking about SOA if it is not related to this paper? There is nothing to say about Operational Research?
>> It seems to me that figures 1, 4, and 5 can have greater legibility, optimizing the use of the space and increasing the size of the letters in texts.
>> I also suggest to include a legend on figures 4 and 5 explaining the numbers in parenthesis, although they are defined in the previous page. Is helps the readers when they are analyzing the image.

Experimental design

>> Keyword Plus by WOS - I did not find any explanation in the text about it. What does “WOS” mean? (all acronyms should be explained). Besides, there is no reference to Keyword Plus. What is the difference to the keywords assigned by the authors method?

Validity of the findings

>> The previous “Limitations” section was removed at all. A section discussing the Threats to Validity is completing missing in the paper.
>> The Conclusion lacks a discussion on the opportunity for future work. Therefore, does not identify unresolved questions / gaps / future directions.
Answer 10: Thanks for the comment, this section has been expanded.
>> Future work analysis is still missing in the new version of the paper.

Additional comments

The main points of the previous review were addressed, and the work has undergone a significant improvement concerning its analyzes.
However, I am still missing a more in-depth related work section and a discussion about Threats to Validity. All works have their limitations, and it is up to the authors to point them out and discuss how they dealt with them in the research.
And as the last point, I would like to suggest again that the work be subjected to an English review, as I identified new errors being introduced in this new version of the work. For instance: “The review business process modelling literature and describe the…” is a phrase with no meaning.

Reviewer 2 ·

Basic reporting

The paper should be re-read, preferably by an English native, as some sentences are weirdly phrased.

For instance, in the conclusion:
"This area (BE) became one of the papers in 2017, and its significant onset and the future impact can be expected."

Not sure what this sentence means... The BE area became one of the papers... Please rephrase and re-read the entire paper to make it ready for publication.

Experimental design

I have no remaining comments regarding study design.

Validity of the findings

While my previous comment on validity of findings was partially addressed, I still expect a deeper discussion on research gaps and future opportunities, specifically in combining BPMN and operational research. Open topics for future research should be identified more clearly.

---

## Round 0.3 · accepted · Accept

I am pleased to inform you that your manuscript has been accepted for publication.  Please find below my comments and any reviewer comments.

Reviewer 1 ·

Basic reporting

I have no remaining comments. The authors addressed my observations.

Experimental design

I have no remaining comments. The authors addressed my observations.

Validity of the findings

I have no remaining comments. The authors addressed my observations.

Additional comments

I have no remaining comments. The authors addressed my observations.

Reviewer 2 ·

Basic reporting

I suggest that the paper undergoes thorough English proof-reading before publication.

Experimental design

No further comments.

Validity of the findings

No further comments.

Additional comments

No further comments.